# Experimental and Mathematical Models for Real-Time Monitoring and Auto Watering Using IoT Architecture

**Jabar H. Yousif * and Khaled Abdalgader**

Faculty of Computing and Information Technology, Sohar University, Sohar 311, Oman; komar@su.edu.om
*   Correspondence: jyosuif@su.edu.om; Tel.: +968-26720101 (ext. 307)

**Abstract:** Manufacturing industries based on Internet of Things (IoT) technologies play an important role in the economic development of intelligent agriculture and watering. Water availability has become a global problem that afflicts many countries, especially in remote and desert areas. An efficient irrigation system is needed for optimizing the amount of water consumption, agriculture monitoring, and reducing energy costs. This paper proposes a real-time monitoring and auto-watering system based on predicting mathematical models that efficiently control the water rate needed. It gives the plant the optimal amount of required water level, which helps to save water. It also ensures interoperability among heterogeneous sensing data streams to support large-scale agricultural analytics. The mathematical model is embedded in the Arduino Integrated Development Environment (IDE) for sensing the soil moisture level and checking whether it is less than the predefined threshold value, then plant watering is performed automatically. The proposed system enhances the watering system's efficiency by reducing the water consumption by more than 70% and increasing production due to irrigation optimization. It also reduces the water and energy consumption amount and decreases the maintenance costs.

**Keywords:** agricultural development; watering management; IoT architecture; predicting models; irrigation performance

## 1. Introduction

The need for efficient management irrigation systems has become crucial in many regions worldwide due to the scarcity of water resources because of the changes in climatic conditions, high atmosphere temperature, and the negative impact of human behavior on the environment. The availability of water has become a global problem affecting many countries, especially in remote and desert areas. Oman is one of the countries with large desert areas that lack potable water sources, and the rise in temperatures also leads to the rapid loss of water from the land. Therefore, there is a need for an efficient irrigation system that works automatically to improve irrigation operations, reduce water consumption, and reduce energy costs. The purpose of watering is to give the plants the right amount of water to ensure ideal growth. Optimal irrigation management aims to determine the timing and quantity of water suitable for irrigation to achieve the most significant effectiveness. Developments in industry tools, information technology, and communication have helped innovate irrigation methods that consume less water than manual and old technologies [1]. Therefore, intelligent irrigation methods lead to less water consumption and reduce the field's excess water, which leads to better crop production [2]. Finding improved techniques that improve the water use efficiency and lower the energy usage has become an affluent research area. Developing an autonomous architecture is considered an ideal approach for processing and analyzing the sensed data for supporting real-time monitoring of agricultural parameters. It also ensures interoperabil-

ity among heterogeneous sensing data streams to support large-scale agricultural analytics [3]. In recent years, precision agriculture has received considerable concern due to the increasing demand for food production with high-quality crops, minimum cost, and reducing the effects of environmental pollution. Wireless sensor network technologies are utilized for providing solutions in the agricultural domain. It aims to provide an optimal tool for collecting, processing, managing, and analyzing the relevant agricultural information and farming activities [4]. The main advantage of these technologies is their ability to create a network of enabled devices (i.e., sensors) that can capture environmental parameters related to agriculture fields and transmit them to the predefined application for further processing and analysis [5]. However, many plantations' attributes such as soil types, fertilizer processes, water requirements, and weather conditions in agriculture fields have different needs and considerations [6]. Many researchers have discussed the need to develop a self-watering mechanism to increase the efficiency of farming systems and reduce the percentage of discharged water.

This work suggested an automatic irrigation method based on a developed mathematical model derived according to the nature of the land and climatic conditions such as temperature and humidity. The proposed model can be easily and quickly changed to meet any changes in climatic conditions.

Additionally, the proposed model helps to manage and monitor plants' needs in an efficient manner. The use of sensors helps to use water efficiently and reduce the water consumption and energy needed for irrigation, reducing the need for labor to turn the motor ON and OFF, controlled by the automated irrigation system based on renewable energy.

Most existing systems require a connection to the Internet and external data storage to manage and control the plant's needs. The proposed method helps manage and control the plants' needs automatically without the need for the Internet. It is embedded in the field and can easily update it for any new conditions.

## 2. Related Work

Many researchers have proposed autonomous methods for watering plants based on mathematical models derived by machine learning methods.

Abrishambaf et al. proposed an autonomous approach to improving the irrigation efficiency based on water needs through field data such as temperature, wind, soil moisture, and soil evapotranspiration estimation. The results show that the proposed approach schedules irrigation efficiently and lowers the cost periods and energy price [7]. Munir et al. suggested a smart watering system (SWS) based on a Fuzzy Logic controller using an Android to optimize water waste in small and medium-scale fields. They deployed a set of sensors based on Blockchain technology that allows trusted devices to capture plants' real-time data and environmental conditions such as soil moisture, humidity, temperature etc. The Fuzzy Logic method is used to control the watering requirements and make the right decisions for turning water tunnels ON/OFF [8]. Similarly, Kolias et al. proposed the GreenIQ Smart Garden system that schedules the watering plan for plants based on the current and historical forecasted weather conditions. The proposed approach provides a friendly user interface that allows users to select weather forecasting services and instruments. The proposed algorithm compensates for lacking the correct weather information. It helps save water accurately by managing the duration of the irrigation cycle, taking into account the weather variables such as the humidity, temperature, wind speed, etc. [9]. Pienaar et al. presented an automated irrigation scheme with a low cost using an impedance moisture sensor method. The proposed Arduino technology is implemented for controlling the irrigation process of the greenhouse. An efficient algorithm is proposed to determine and optimize the water level by comparing the plant environmental data such as the temperature, humidity, and soil level with the statistical results [10].

Harun et al. introduced an enhanced indoor farming IoT monitoring system for remote monitoring the growth of the Brassica Chinensis plant. Light sensors were used to

monitor the spectrum from a distance and sensors to measure carbon dioxide content, ambient temperature, humidity, and leaf area index. The watering method is controlled by the pulse-width modulated (PWM) actuator using an IoT embedded device. The study demonstrated the light spectrum and intensity effect on Brassica Chinensis in determining the optimal plant physiology and morphology, such as water use efficiency, leaf photosynthesis, and chlorophyll rate [11]. Ashton K. [12] suggested the interconnection method between different devices using the Internet of things (IoT), which provided a facility for sensing, processing, and analyzing environmental information. These devices are commonly using standard network protocols to perform intercommunication with each other. Ashton's approach aimed to develop devices that self-generated reports in a real-time manner for enhancing the efficiency and accumulating relevant information. Ofrim et al. [13] used the ZigBee wireless sensor network for developing an automating irrigation system for managing irrigation timing and watering needs in different soil moisture conditions. The irrigation process is considered one of the essential issues in the agriculture domain, where different irrigation approaches are used for managing water wastage in conventional irrigation methods. Damas et al. [14] proposed a remote-controlled water irrigation system for several agricultural regions. They used computer networks to connect all the areas with the central controller to automate the irrigation process. The empirical results have shown that the proposed system saved up to 30–60% of the consumed water. In addition, the method proposed by Evans and Bergman [15] controlled the irrigation process by using wireless sensors to collect the surrounding environmental information to help produce an irrigation schedule.

Various sensor-based systems have been proposed to help control irrigation water and improve the utilization of water resources and production. Basu et al. [16] presented an automatic irrigation control system based on sensors for sensing environmental-related agriculture parameters and storing the sensed information for further statistical analysis. Kim et al. [17] also proposed an irrigation system that remotely monitors environmental parameters such as soil moisture using GPS and Bluetooth technologies. They deployed a sensor-based system that helps to increase the productivity of the crop and reduce water consumption. Kim and Evans [18] developed a site-specific sprinkler irrigation system using remote sensors based on Bluetooth wireless radio communication. They integrated a site-specific controller to support real-time decision-making on irrigation processes. Using wireless sensors in the agriculture domain is currently the focus area of research. Fourati et al. [19] proposed a wireless sensors system to measure environmental attributes such as humidity, temperature, and solar radiation to develop a web-based decision support system that provides irrigation scheduling in agriculture fields.

Kaewmard and Saiyod [20] also proposed an automation agriculture approach based on long-term sustainability. The connected sensors can be moved in the vegetable fields to record all possible changes in the environmental parameters. Hashim et al. [21] developed an Arduino-based system for measuring and monitoring soil moisture and temperature parameters through a smartphone application. They compared the advantages of small-scale and large-scale agriculture-related architectures. It claimed that small-scale systems do not cost as much as large-scale systems that require expensive components. Srbinovska [22] proposed another aspect of real-time monitoring in the agricultural fields to improve the quality of products using wireless sensor network architecture. They are focused on the faulty tolerance and energy efficiency of employed sensors in sensing agriculture-related parameters.

Nawandar et al. [23] proposed a low-cost intelligent irrigation system using a neural network method for determining the sensor input based on the irrigation schedule for efficient irrigation. The proposed devices offered several facilities, such as irrigation schedule estimation, decision making, and remote data monitoring. Sarkar et al. [24] developed a virtual sensing framework (VSF), which helped to reduce the network's data traffic and transmission. They deployed a cross-correlation method for predicting multiple consecutive sensed data and achieved an accuracy of 98%. Benyezza et al. [25]

developed an automated irrigation embedded system based on Arduino for optimizing water use and monitoring the field.

The analysis of the literature survey indicates the gaps that need to be addressed:

- There is no suitable control and management model that determines the water level needed for irrigation.
- Provide the scalability, privacy, and reliability of sensing data using cloud computing and Blockchain technologies.
- Lack of a customizable model that can determine the water conditions based on the type of plant, even in the same soil and weather conditions.
- Lack of correct weather information such as humidity, temperature, and wind speed is used accurately to determine the level of needed water and manage the irrigation cycle's duration.

On the other hand, there is a need for an automatic irrigation model with the following features:

- Simple and easy to install and configure.
- Save energy and time to water at the correct time, utilizing anywhere with less effort.
- Use the needed amount of water and reduce the amount of overwatering to improve the crop performance.
- Reduce the need for labor to turn the motor ON and OFF, controlled by the automated irrigation system.
- Reduce human error elimination in adjusting available soil moisture levels.
- This manuscript introduces an IoT embedded system for an auto-watering and real-time monitoring approach to improving the efficiency of irrigation needs based on mathematical models that determine the plantations' water requirement.

## 3. Materials and Methods

This section describes the proposed IoT architecture based on experimental and mathematical models for auto watering and real-time monitoring of heterogeneous sensing agricultural parameters.

This manuscript deployed a model-based design (MBD) and experimental research methods for developing an embedded automatic irrigation control system. The MBD performs verification and validation by testing the proposed mathematical model and algorithms developed to control the Arduino microcontroller, sensors, running motor, pump, and solar energy. The experimental design ensures that the proposed model controls and monitors the automated irrigation system to obtain feedback from sensors, water levels, and activate the watering motor automatically ON/OFF.

Figure 1 shows the main components of the proposed architecture.

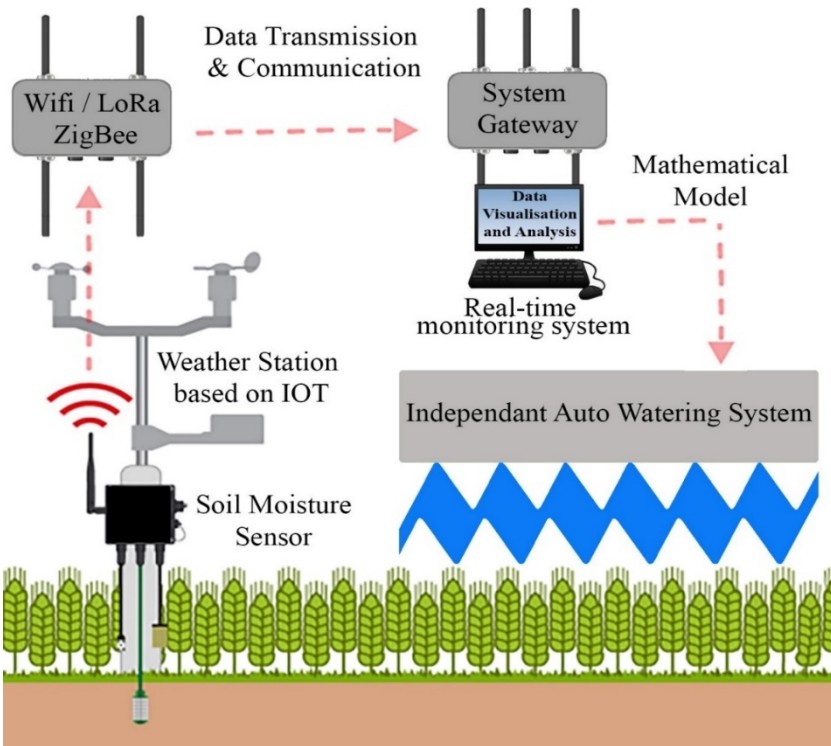

**Figure 1.** The main components of the proposed architecture.

### 3.1. System Set-Up and Instalation

The proposed experimental system was installed, tested, and evaluated in Sohar city, Oman—located at latitude: 24°21′0.79″ N, longitude: 56°42′27.54″ E—to evaluate its performance and effectiveness. The proposed agriculture-related architecture is implemented using a Libelium Smart Agriculture Vertical Kit, including various agricultural-related sensors [26]. This architecture consists of five layers: data source layer, data collection layer, data transmission layer, data processing layer, and data viewing layer. Unlike existing cloud-based architectures, whereas the connection to the cloud platform is essential for receiving analyzed information, the proposed architecture allows the farmers to remotely measure and monitor the agricultural parameters in real time directly via wireless communication technologies. The data source layer is responsible for sensing agricultural parameters (data) using different types of sensors.

These sensors can be installed in the soil and the surrounding environment. Soil sensors are mainly water-resistant and usually sensing parameters related to soil moisture, temperature, and other soil properties. Surrounding environment sensors, however, measure environmental parameters such as air temperature, air humidity, atmospheric pressure, rain level, wind speed and direction, solar radiation, and leaf wetness [27].

The sensors are connected directly with a sensor node consisting of a wireless antenna, the ports panel for interfacing with the sensors, and a built-in solar energy source, as shown in Figure 2. This experiment configured each sensor node to send a frame of collected data to the data processing layer approximately every 15 min through the LoRa communication channel. This is because it helps reduce the power consumption and save its associated charged battery using an external solar panel. The installed weather station used three types of wireless communication technologies to connect the data collection and processing layers: LoRa, WIFI, and ZigBee. LoRa is used to achieve long-range connections, and Wi-Fi provides a decent communication range up to 100 m with a data transmission rate of 2 to 54 Mbps at 2.4 GHz radiofrequency. ZigBee is a short-range radio communication technology used for transmitting data frames over long distances using LoRa technology.

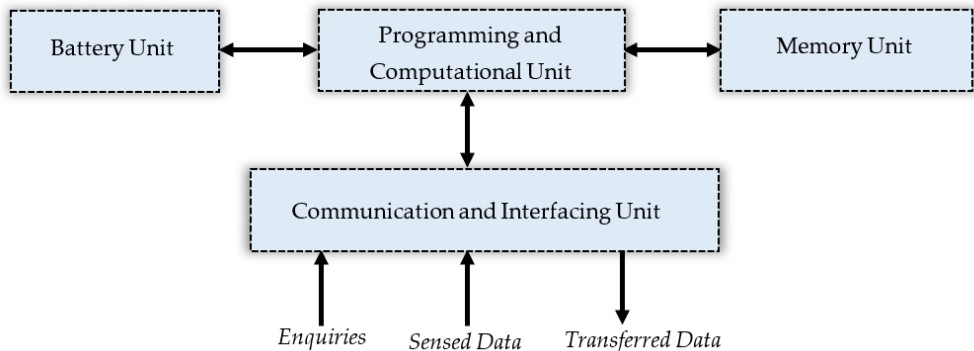

**Figure 2.** Sensor node architecture.

Table 1 summarizes the communication technologies that are used in the proposed agriculture-related architecture.

**Table 1.** A comparative of the used communication technologies.

| Communication Technology | Data Rate (Bandwidth) | Transmission Range | Operating Frequency |
|---|---|---|---|
| LoRa | 0.3–50 Kbps | 2–5 km | 433,868,780,915 MHz |
| WiFi | 2–54 Mbps | 20–100 m | 2.4 GHz |
| ZigBee | 20–250 Kbps | 10–20 m | 868/915 MHz, 2.4 GHz |

*3.2. Experemintal Set-Up and Instalation*

This experiment's agricultural system and its components are all based on the flow chart in Figure 3. Firstly, the weather station is installed. The scenario involves two weather station nodes, each in different plant pots (5 kg soil per pot). The experiments are made using two sensor nodes, as depicted in Table 2.

The amount of water that is given in each interval is half a liter (0.5 L). The installed components of the proposed system are shown in Figure 4. The first sensor node (node 1—Figure 4a) was watered based on the information captured by the associated moisture sensor that shows the need for water. The second node (node 2—Figure 4b) was watered manually once per day when needed.

**Table 2. The** specifications of installed sensor in nodes 1 and 2.

| Sensor Node 1 | Sensor Node 2 |
|---|---|
| Temperature, humidity, and pressure probe | Temperature, humidity, and pressure probe |
| Soil moisture 30 cm probe | Soil moisture 30 cm probe |
| Soil moisture 10 cm probe | Solar radiation probe |
| Soil/water temperature (Pt-1000) probe | Soil/water temperature (Pt-1000) probe |
| Leaf wetness probe | |
| WS-3000 (anemometer, wind-vane, pluviometry) probe | Leaf wetness probe |

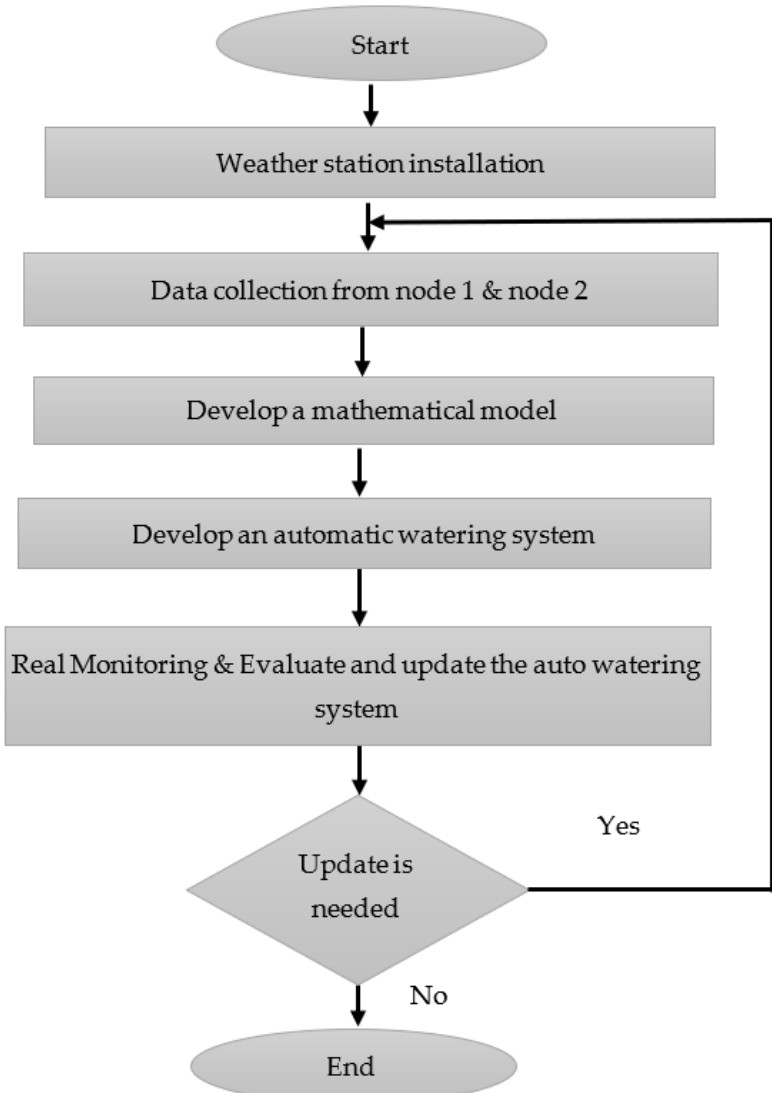

**Figure 3.** The main components of the proposed architecture.

Data relating to the plant's environment, such as the temperature, humidity, pressure, wind speed, and soil moisture, were gathered using the weather station for 51 days. This weather station uses the LoRa radio (XBee protocol) under the frequency of 2.4 GHz for communication between sensors node and system gateway (Figure 4c). A receiver (system gateway) is connected to the user's computer to receive and access the required data. Then, these data were used to create mathematical models that can be used to accurately and efficiently predict the plant's environmental requirements for future use. The values that are generated from these mathematical models are then implemented in the independent auto-watering system. Additionally, the weather station continues sending the recorded data, which is used for real-time monitoring of the environmental conditions of the two plants. In case of any irregular situation, a proper solution needs to be taken to solve this issue.

The amount of irrigated water in each required irrigation time is half a liter for both plant pots. The two plant pots were placed in the exact location under the same weather condition during the experimental process, taking 51 days of data recording (i.e., started on 24 April 2019, where the total sensed data were 4893 XBee frames). Figure 5 shows the environmental parameters (e.g., temperature "TC" and humidity "Hum") around the installed plant pots that are located and irrigated under the same weather condition. The

Hum_ node 2 is the humidity values, and TC_ node 2 is the Centigrade temperature values (°C) recorded by the Sensor Node 2. Additionally, Hum_ node 1 is the humidity values, and TC_ node 1 is the Centigrade temperature values recorded by Sensor Node 1. Figure 5b shows the daily amount of irrigated water for each plant in the experimental time in the two pots.

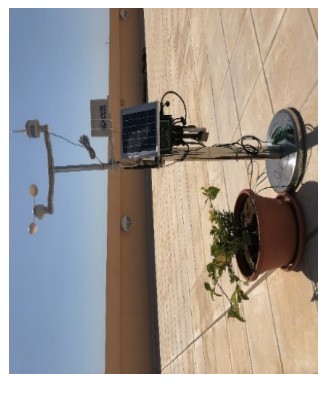 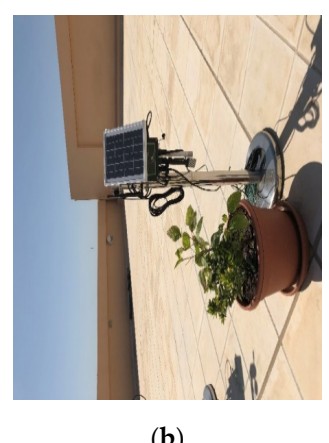 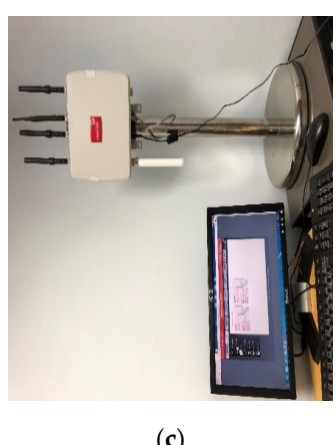

| (**a**) | (**b**) | (**c**) |

**Figure 4.** Installed components of the proposed system. (**a**) Sensor Node 1; (**b**) Sensor Node 2; (**c**) System Gateway.

Several statistical methods for describing statistics analysis results include minimum and maximum values, median, different quartile status, mean, variance, and standard deviation. Table 3 presents the descriptive analysis information regarding the soil moisture (Hz) parameters for both installed plant pots (node 1, node 2). Soil_C_ node 2 represents the soil moisture information of the plant under node 2, whereas Soil_C_ node 1 represents the soil moisture information of the plant under node 1.

The first quartile (Q1) is the middle value between the minimum amount and the median of the dataset. The third quartile (Q3) is the central value between the median and the maximum number of the dataset. The results show that the minimum and maximum statistical values for plants under sensor nodes node 2 and node 1 are 0, 20.25, 111.11, and 51.68, respectively. It can also be seen that the median statistical value for plants under sensor nodes node 2 is 87.71 and 32.87 for the plant under sensor node 1.

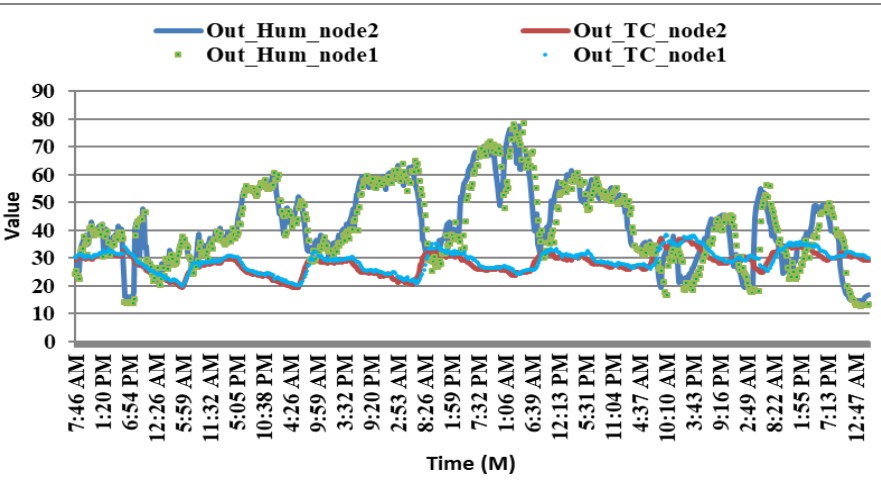

(**a**)

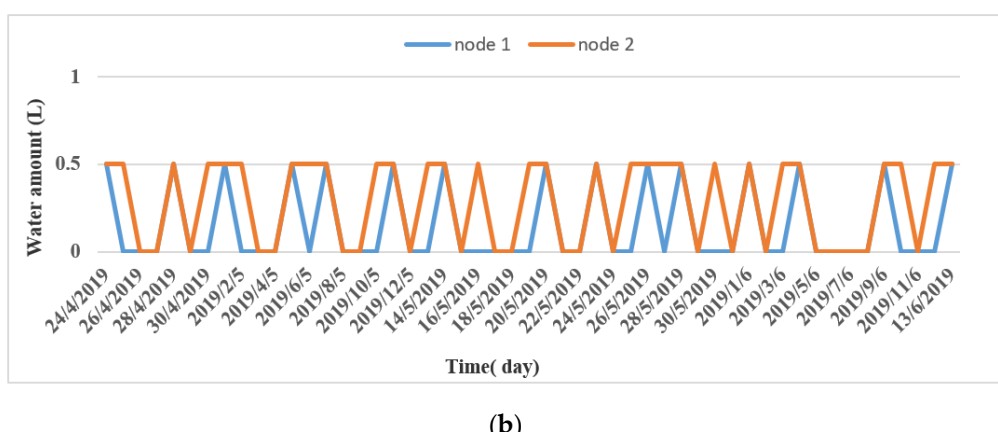

(**b**)

**Figure 5.** (**a**) Environment condition of the two installed plant pots; (**b**) amount of water and irrigation time for sensor node 1 and node 2.

**Table 3.** Descriptive statistics using several statistical analysis methods.

| Statistic | Soil_C_ Node 2 | Soil_C_ Node 1 | Hum_ Node 2 | TC_ Node 2 | Hum_ Node 1 | TC_ Node 1 |
|---|---|---|---|---|---|---|
| No. of observations (stored frames) | 4893 | 4893 | 4893 | 4893 | 4893 | 4893 |
| Minimum | 0.0000 | 20.25 | 7.00 | 19.20 | 3.91 | 20.01 |
| Maximum | 111.11 | 51.68 | 99.60 | 45.03 | 100.00 | 47.57 |
| 1st Quartile | 64.10 | 28.87 | 29.51 | 28.99 | 28.09 | 29.64 |
| Median | 87.71 | 32.87 | 42.61 | 31.91 | 40.85 | 32.78 |
| 3rd Quartile | 96.15 | 38.18 | 60.41 | 35.24 | 59.02 | 36.20 |
| Mean | 78.59 | 33.69 | 46.18 | 32.17 | 45.22 | 33.03 |
| Variance (n-1) | 590.72 | 34.86 | 424.14 | 20.00 | 474.14 | 21.74 |
| Standard deviation (n-1) | 24.30 | 5.90 | 20.59 | 4.47 | 21.77 | 4.66 |

*3.3. Proposed Auto-Watering System*

The proposed auto-watering system is an independent auto-watering system controlled by Arduino IDE, which is used to write the code and for all of the testing data. The sensed data collected from the weather station is used to predicate the mathematical models. The flowchart of the embedded Arduino IDE model that manages and monitors the auto-watering system is presented in Figure 6. The soil moisture content can be presented in the percent of volume as in Equation (1).

$$\text{soil moisture content (SMC)} = \frac{\text{Depth m}^3}{\text{Volume m}^3} \times 100 \tag{1}$$

In this experiment, the soil depth is 0.5 m³, the volume is 1 m³, and the soil amount is 50%. Several depth readings can be obtained by using a multi-depth soil moisture probe. The first sensor can be at (10 cm) below the surface; additional sensors should be installed at (25–30 cm). The standard level of the water amount is determined significantly based on the soil texture and structure, as presented in Figure 7.

The proposed auto-watering system shown in Figure 8 has a negative feedback loop to keep the soil moisture at acceptable levels. When the soil moisture is below a certain threshold, the Arduino automatically activates the valve and lets the water pour into the soil. Once the soil moisture reaches an acceptable level, the valve is deactivated. The system is based on a soil moisture sensor planted inside the soil to monitor the water levels every 15 min. Based on the soil moisture value, a 12 V solenoid valve was used to control the water flow in the soil automatically. An external 12 V power supply (solar panel) is used to power the valve and the Arduino (using a DC-DC 5 V converter). A relay (along

with the external power supply) connects the Arduino and the valve. During testing, to ensure no irregularities in the plant's environmental conditions, arching occurred when the wires were connected directly, which caused the relay's burning. A flyback diode was implemented parallel to the relay and the valve, so the relay was used without any problems. Other types of equipment such as a basic breadboard and jumper wires were used to connect everything. The weather station will continue to operate alongside the auto-watering system to ensure no irregularities in the plant's environmental conditions. If there were, a signal could be sent to the user, alerting them of any changes to carry out appropriate actions to the auto-watering system and maintain its efficiency.

```
#include <dht.h>
#define dht_apin A0
dht DHT;
void setup() {
   // put your setup code here, to run once:
   Serial.begin(9600);
   pinMode(7, OUTPUT); }
void loop() {
   // put your main code here, to run repeatedly:
   DHT.read11(dht_apin);
   if (DHT.humidity < 25)
      {digitalWrite(7, HIGH); }
   if (DHT.humidity > 50)
      {digitalWrite(7, LOW); }
      delay(900000);   }
```

**Figure 6.** The embedded Arduino code to control the auto-watering system.

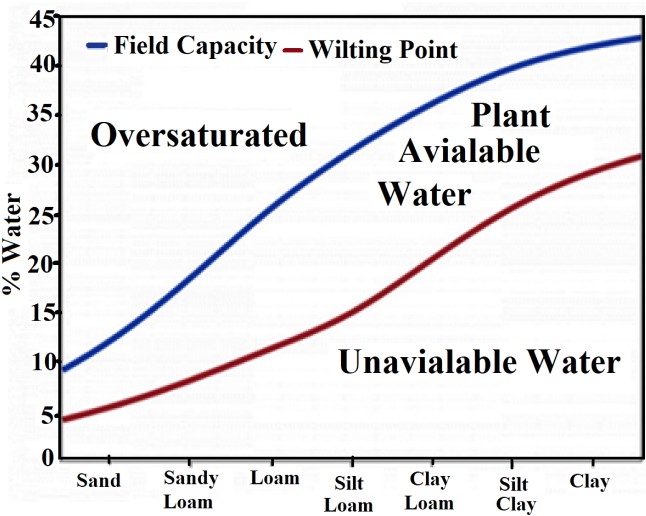

**Figure 7.** The water content percent based on the soil texture.

**Figure 8.** The proposed auto-watering system based mathematical model; red line is positive terminal; black line is ground; yellow line is digital data; blue is analogue data.

## 4. Results and Discussion

This section reviews the obtained results and discusses the main contribution of the proposed experimental architecture and mathematical models' implementation.

### 4.1. Performance Evaluation Measures

Several standard performance evaluation metrics were proposed to evaluate the accuracy of prediction results, such as the coefficient of determination (R2), mean squared error (MSE), root mean square error (RMSE), mean absolute percentage error (MAPE), and mean absolute error (MAE) [28].

The coefficient of determination (R2) is considered one of the important measures for verifying the performance of predicting models, which has an approximate value from 0 to 1. The closest ($R^2$) value to 1 is indicated as the best performance result, and it can be defined as in Equation (2):

$$R^2 = 1 - \frac{\sum_i^n (y_i - f_i)^2}{\sum_i^n (y_i - \bar{y}_i)^2} \tag{2}$$

where $y_i$ is the experimental data and $\bar{y}_i$ is the mean of the experimental data. $f_i$ is the predicted data of $y_i$ and n is the sample size. In some cases, the coefficient of determination is misleading when its value is negative, confusing with a squared letter with negative values. Therefore, the adjusted R-squared is used for examining the performance of predicting data, in which its value is increased if extra variables are involved in the model. The adjusted R-squared is computed as in Equation (3):

where n is the sample size and k is number of variables in the model.

The mean squared error (MSE) is the average cost (i.e., squared difference) between real value and the obtained values, which is calculated as in Equation (4):

$$R^2{}_{adj} = 1 - \left[\frac{(1 - R^2)(n - 1)}{(n - k - 1)}\right] \tag{3}$$

$$MSE = \frac{\sum_{j-0}^{p} \sum_{i-0}^{n} (d_{ij} - y_{ij})^2}{np} \tag{4}$$

where $p$ is the number of processing elements, n is the sample size, $y_{ij}$ is processing output exemplar (i) at processing element (j) and $d_{ij}$ is the experimental output for exemplar (i) at processing element (j).

Another related metric that has been used in our experiments to evaluate the obtained results is the root mean square error (RMSE), as defined in Equation (5):

$$RMSE = \sqrt{\frac{1}{N} + \sum_{i=1}^{N} (y_i - f_i)^2} \tag{5}$$

### 4.2. Proposed Mathematical Models

Both plants were in good condition during the experimental time between 24 April 2019 and 13 June 2019. The total amount of water consumed under sensor node 1 (watering on demand) was 7.5 L, whereas the plant under sensor node 2 (watering manually) consumed 14.5 L of water. This indicates that half of the irrigated water was wasted at the plant irrigated daily (node 2). Therefore, the experiment proves that wasting water can be reduced by improving the efficiency of collecting important sensed information in a proper automatic monitoring system. Moreover, the moisture level of node 2 reaches a high percentage of 85–90%, which could negatively affect the plant's growth, while the level of moisture of node 1 is about 25–35%, as shown in Figure 9. The good conditions of the two plants indicate that the adequate soil moisture level is between 25% and 35%.

The classification and regression trees methods are used to determine the conditions of both, which devised two clusters according to the following two rules:

R1: If S_ node 1 in [0, 0.25) then node 1 = 0

R2: If S_ node 1 in [0.25, 0.5] then node 1 = 0.5

R1 indicates that if the soil moisture has a value less than 0.25, then watering is required.

R2 indicates that if the soil moisture has a value between 0.25 and 0.5, then watering is not required.

These two rules are used to control the auto-watering system, which is embedded in Arduino IDE. The regression technique is used to analyze the relationships between a set of independent and dependent variables. The regression equation contains several coefficients that explain the relationship between each independent variable and the dependent variable, which enables the prediction of future values. Several types of regression are introduced, mainly distributed into linear and nonlinear techniques. Most of them construct linear regression estimates between X and Y as Y = XB + B0, X is the rank of the matrix, and the algorithm will yield the least-squares regression estimates for B and B0.

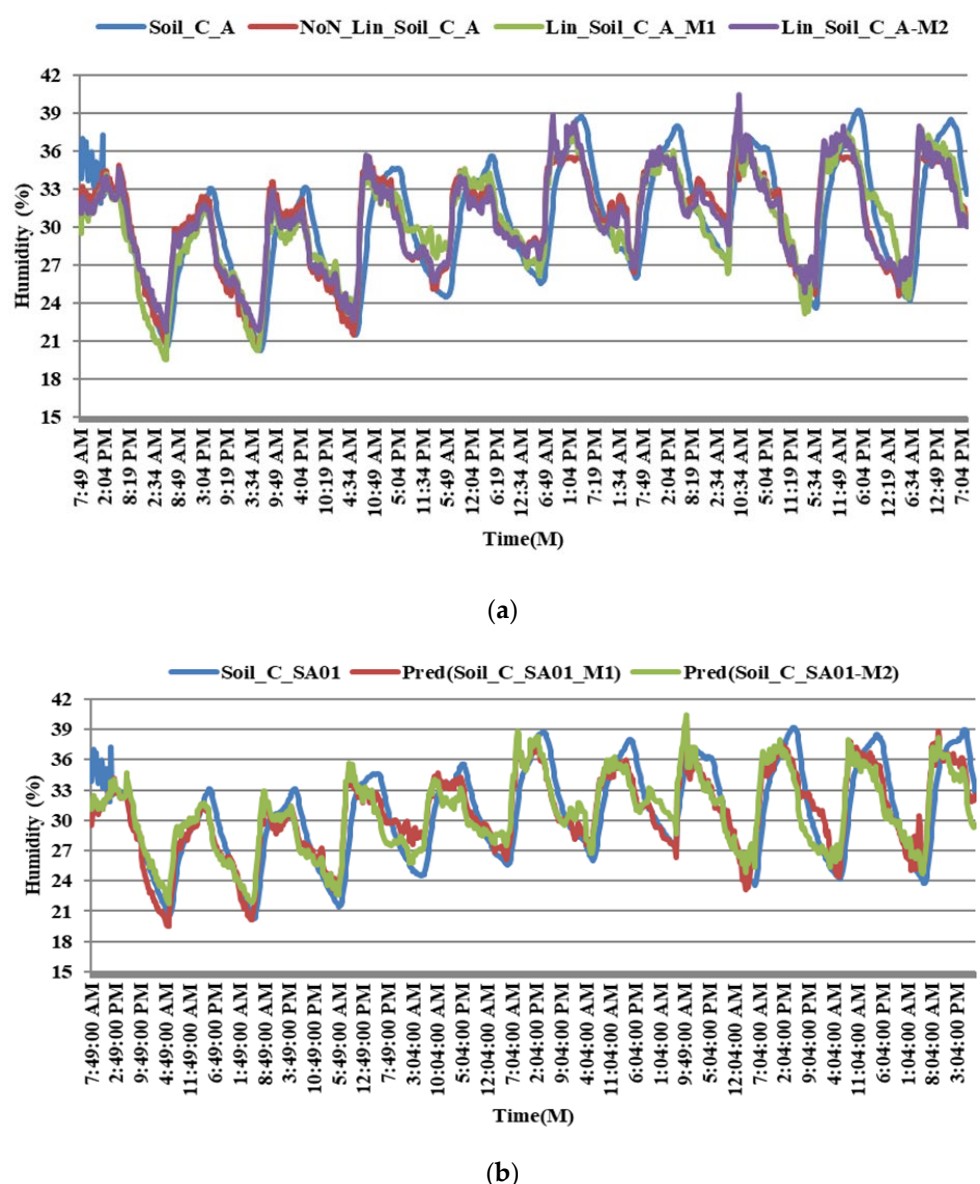

(**a**)

(**b**)

**Figure 9.** Comparison of experimental results and proposed mathematical models (Model 1, Model 2). (**a**) Simulation output; (**b**) predicting output.

Figure 10a presents the soil humidity level based on the two irrigation methods (node 1 and node 2). Figure 10b shows the proposed mathematical model results for controlling the irrigate amount compared to the previous two methods (node 1 and node 2). It indicates that the watering amount is reduced to a quarter of a liter, which reduces the amount of water needed.

Additionally, a linear regression method (Lin-M0) is proposed to determine the amount of soil humidity obtained by Equation (6).

$$\text{Prop (soil-node 1)} = 0.51724 * \text{soil-node 2} \tag{6}$$

Sensitivity analysis helps to analyze the effect of different values of a set of independent variables on a particular dependent variable under specific settings. The results of the sensitivity analysis proved that the temperature (TC) and humidity (Hum) are the variables with the highest impact on the level of soil moisture (soil_C). Table 4 presents the correlation relationship between the dependent variables (temperature (TC_ node1), humidity (Hum_ node1), and an independent variable (soil moisture (soil _C_ node1)) using

automatic irrigation (sensor node 1). It indicates that the soil moisture is affected more by the temperature based on the positive relationship of (0.805). At the same time, the humidity parameter has adverse effects on soil moisture results based on the negative correlation relationship of (−0.5039).

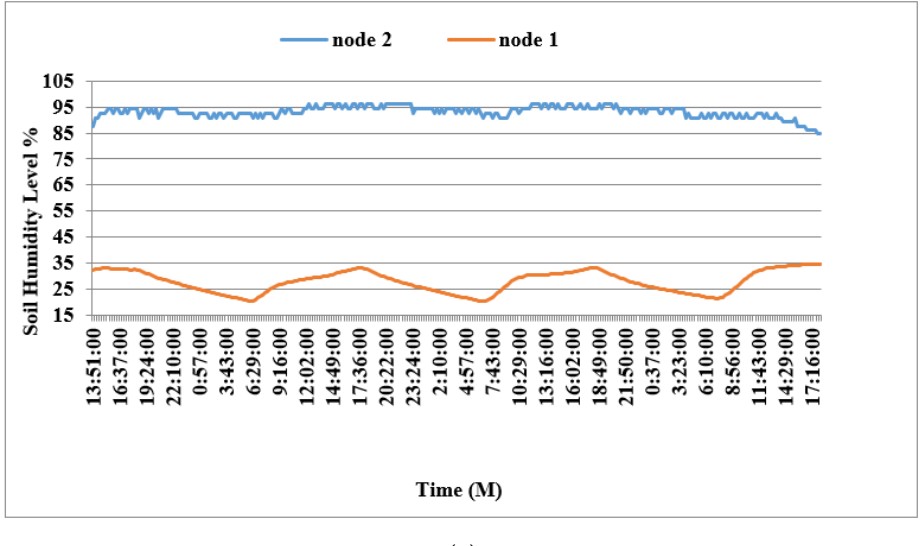

(**a**)

(**b**)

**Figure 10** (**a**). Soil humidity level of the two irrigation methods (node 1 and node 2); (**b**) water amount model (node 1)—Equation (6).

For a real-time monitoring system, we should follow and check the sensed data from the weather station (temperature, humidity) to determine any irregulating environmental conditions. So, it will send an alert message in case of irregulating conditions. Additionally, for the sake of a real-time monitoring system, two models were proposed to predicate the level of humidity level in node 1, which will be used to compare with online sensed data from the weather station to alert when irregulated data is obtained from the weather station (low or high humidity).

**Table 4.** Correlation relationship between the model variables.

| Variables | TC_ Node 1 | Hum_ Node 1 | soil_C_ Node 1 |
|---|---|---|---|
| TC_ node 1 | 1.0000 | −0.5039 | 0.8053 |
| Hum_ node 1 | −0.5039 | 1.0000 | −0.1547 |
| Soil_C_ node 1 | 0.8053 | −0.1547 | 1.0000 |

*4.3. Mathematical Regression Models*

Three mathematical regression models (linear and nonlinear) were developed to analyze and monitor the current values' behavior and predict future conditions. The temperature (TC_ SA01) and humidity (Hum_ SA01) were used as input variables for predicting the future requirements of the soil moisture (soil_C_ SA01). Model 1 used two input variables—temperature (TC_ SA01) and humidity (Hum_ SA01)—whilst Model 2 used only one input variable—the temperature (TC_ SA01). Model 3 is a nonlinear regression model that uses temperature (TC_ SA01) as an input variable, as shown in Figure 10a. The following are the three models as defined in Equations (7)–(9):

Model 1: linear regression.

$$(Lin\_soil\_C\_ SA01) = -6.52 + 1.11446* \text{ temperature} + 0.09873* \text{ humidity} \tag{7}$$

Model 2: linear regression.

$$(Lin\_soil\_C\_ SA01) = 3.34 + 0.92065* \text{ temperature} \tag{8}$$

Model 3: nonlinear regression.

$$(NoN\_Lin\_soil\_C\_ SA01) = 63.63 - 6.72748* \text{ temperature} + 0.31339* \text{ temperature}^2 - 0.00434* \text{ temperature}^3 + 6.25652E-6* \text{ temperature}^4 \tag{9}$$

The goodness of fit statistics of the proposed models are presented in Table 5. The coefficient of determination ($R^2$) values of the three models is (0.73, 0.64, and 0.67), respectively, which means the predicted values are a 70% close fit to the experimental datasets.

**Table 5.** Goodness of fit statistics of the proposed models.

| | Model 1 | Model 2 | Model 3 |
|---|---|---|---|
| $R^2$ | 0.73 | 0.64 | 0.65 |
| $R^2$_adj | 0.73 | 0.64 | 0.65 |
| MSE | 5.97 | 7.85 | 0.132 |
| RMSE | 2.44 | 2.80 | 0.363 |

The value of the adjusted R-squared is precisely equal to the coefficient of determination ($R^2$), which means that the predicted values are in the correct direction of experimental data. The mean squared errors are less in Model 1 (5.97) than in Models 2 and 3 (7.85 and 7.24), respectively. Figure 10b shows the graph of experimental datasets and the predicting datasets of proposed models (Model 1 and Model 2).

The predicting datasets fit the experimental datasets smoothly, which means that we could examine the behavior of the independent variable soil moisture (Soil_C_ SA01) quickly and at a low cost. This will help farmers and decision-makers evaluate the required water rate for a specific time (monthly, annually, every 5 years, etc.). Comparing the results with other researchers is one of the important ways to verify the effectiveness of the extracted results compared to other studies. The results should be compared under the same conditions to give credibility to the results extracted. However, one of the most significant obstacles that we face when carrying out the comparative study is the disparity of the work environment and the conditions of input and outputs. Therefore, we often try to find some common parameters and compare them based on them.

Table 6 presents the results of the proposed methods compared to some studies, which show that most of the proposed methods and these studies have achieved high accuracy. They significantly reduce the percentage of water and energy consumption, which indicates the success of the current experiments. The proposed models reduced the water consumption by about 50–65% compared to 30–60% in [14] and 12.5% in [23]. In addition, another study [24] reduced the energy consumption by up to 69% compared with the proposed models that reduce the energy by using a solar panel to charge the battery.

**Table 6.** The comparison results of the proposed methods with other studies.

|  | Model 1 | Model 2 | Model 3 | [23] | [14] | [24] |
|---|---|---|---|---|---|---|
| R² | 0.73 | 0.64 | 0.65 | 0.98 | - | - |
| R²_adj | 0.73 | 0.64 | 0.65 | - | - | - |
| MSE | 5.97 | 7.85 | 0.132 | 0.06 | - | 0.22 |
| RMSE | 2.44 | 2.80 | 0.363 | 0.77 | - | 0.47 |
| Saving Water | 50–75% | 50–75% | 50–75% | 12.5% | 30–60% | - |
| Saving Energy | Yes | Yes | Yes | - | - | 69% |

## 5. Conclusions

This paper proposed an autonomous sensor-enabled architecture using different self-powered wireless sensors that support real-time monitoring of agricultural parameters over various heterogeneous sensing data streams. The proposed architecture allows the farmers to measure and monitor their farms remotely without a need to access third-party platforms. The architecture is tested and evaluated using real scenarios encompassing the various aspects of the precision agriculture process. The empirical results show that the proposed architecture can be used in a variety of agricultural activities, including the control of irrigation water and the monitoring of agrarian conditions. Sensing and monitoring soil moisture play a significant role in the agriculture domain for assisting farmers in controlling and managing their irrigation methods more efficiently.

The empirical experiments proved that the proposed architecture could efficiently control and monitor the agricultural conditions, minimize water waste, and maximize the growth rates of the plants. Therefore, developing an automatic-sensor-enabled architecture system provides a potential solution for managing the farm accurately. The proposed approach helps maintain the irrigation effectively, uses suitable amounts of water, and enhances productivity. In addition, three mathematical regression models were developed to predict the agricultural activities' future behavior under specific conditions and scenarios.

The main contributions of this work are:

- A critical survey and empirical study conducted to analyze the impact of implementing an autonomous sensor-enabled architecture in Oman to reduce consumed water consumption in irrigation and enhance plants productivity.
- The proposed method helps to manage and monitor plant needs in an efficient manner. The use of sensors helps to control more than one field at a time.
- Most of the existing systems used in managing and controlling plants require a connection to the Internet and external data storage. The proposed method helps manage and control the plants' needs automatically without the need for the Internet.
- The proposed method uses an internal wireless network covering several adjacent fields, which reduces the expenses needed to manage the farms.
- The proposed method works without an energy source, as it generates the energy needed for self-operation by solar panels. It can also work in distant areas where there is no power source. It proposed three mathematical models that simulate irrigation time and plant needs. Moreover, they can predict the amount of water needed for irrigation at any time.

The validity and effectiveness of the proposed methods have been tested mathematically, as well as their conformity with the actual data. However, certain restrictions on the work presented in this paper need to be addressed to improve the proposed architecture's effectiveness. The proposed architecture needs to be implemented in a large-scale field, which will allow the analysis of the impacts of the different weather conditions on the irrigation process in Oman.

**Author Contributions:** Conceptualization, J.H.Y. and K.A.; methodology, J.H.Y. and K.A.; software, J.H.Y. and K.A.; validation, J.H.Y. and K.A.; formal analysis, J.H.Y. and K.A.; data curation, K.A.; writing—original draft preparation, J.H.Y. and K.A.; writing—review and editing, J.H.Y. and K.A.; visualization, J.H.Y. All authors have read and agreed to the published version of the manuscript.

**Funding:** This research received no external funding.

**Institutional Review Board Statement:** Not applicable.

**Informed Consent Statement:** Not applicable.

**Data Availability Statement:** Data available on request due to restrictions.

**Conflicts of Interest:** The authors declare no conflicts of interest.

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
