# Peer review of "Experimental and Mathematical Models for Real-Time Monitoring and Auto Watering Using IoT Architecture"

_computers, doi:10.3390/computers11010007_

Round 1
Reviewer 1 Report
This article deals with a smart irrigation system. However, the article does not qualify as a research article in its current form. It even lacks the required details of a technical report.
*) In the introduction of the article, a number of high-level research gap is mentioned based on literature survey. But two important points of any research article are completely missing.
what is the specific problem statement of this article?
what are the contributions of this article?
Additionally, the introduction should also include what is the approach (high-level) this article followed to solve the problem it is tackling/focusing on?
*) The five layers mentioned in section 2.1 are not shown in fig 1. Is the layered architecture bundled in the off-the-shelf agricultural kit?
*) The three wireless communication protocols mentioned -- are all of them used in the deployment? If yes, why all of them simultaneously? Please justify. If not, what exactly is used in the deployment?
*) In general, this research area can be focused on three aspects -- communication protocol design, data analytics and architecture design that facilitates both efficient communication and data analytics. Please concentrate on one.
*) You have missed a number of relevant works. Please position your work by seeking help from the existing literature, e.g.,
Nawandar et al., "IoT based low cost and intelligent module for smart irrigation system",
Benyezza et al., "Smart Irrigation System Based Thingspeak and Arduino"
Krishnan et al., "Fuzzy Logic based Smart Irrigation System using Internet of Things"
*) For prediction, there exists a number of existing methods as well, e.g.,
Sarkar et al., "VSF: An Energy-Efficient Sensing Framework using Virtual Sensors". Please position your proposal accordingly.
*) Some attention should also be paid to written English.
Author Response
Dear Editor
Dear respected reviewers
I have revised my article according to the editors' and reviewers' comments without any exception. However, below is my detailed response to the received comments. I would like to take this opportunity to thank the editorial office members and the reviewers for their valued comments and feedback, which enriched the article.
Note: the remarks of the reviewers are colored black, while the authors' respond is highlighted in blue
Best regards
Reviewer 1:
This article deals with a smart irrigation system. However, the article does not qualify as a research article in its current form. It even lacks the required details of a technical report.
I would like to take this opportunity to thank the reviewer for the valued comments and feedback, which enriched the article.
Thank you for the helpful comment. The manuscript is enhanced according to the reviewer’s comment. Please refer to the new version.
*) In the introduction of the article, a number of high-level research gap is mentioned based on literature survey. But two important points of any research article are completely missing.
what is the specific problem statement of this article?
Thank you for the helpful comment. Done, the manuscript is updated based on the reviewer’s comment. The problem statement is clearly mentioned in the introduction as follows:
Availability of water has become a global problem affecting many countries, especially in remote and desert areas. Oman is one of the countries with large desert areas that lack potable water sources, and the rise in temperatures also leads to the rapid loss of water from the land. Therefore, there is a need for an efficient irrigation system that works automatically to improve irrigation operations, reduce water consumption and reduce energy costs.
what are the contributions of this article?
Thank you for the helpful comment. Done, the manuscript is updated based on the reviewer’s comment. The contributions of this work are:
This work suggested an automatic irrigation method based on a developed mathematical model derived according to the nature of the land and climatic conditions such as temperature and humidity. The proposed model can be easily and quickly changed to meet any changes in climatic conditions.
Also, the proposed model helps to manage and monitor plants needs in an efficient manner. The use of sensors helps to use water efficiently and reduce water-consuming and needed energy for irrigation. Reduce the need for labor to turn the motor ON and OFF, controlled by the automated irrigation system based on renewable energy.
Most existing systems require a connection to the Internet and external data storage to manage and control the plant's needs. The proposed method helps manage and control the plants' needs automatically without the need for the Internet. It is embedded in the field and can easily update it for any new conditions.
Additionally, the introduction should also include what is the approach (high-level) this article followed to solve the problem it is tackling/focusing on?
Thank you for the helpful comment. Done, the manuscript is updated based on the reviewer’s comment. The research methodology deployed in this manuscript is as the following:
This manuscript deployed a Model-based design (MBD) and experimental research methods for developing an embedded automatic irrigation control system. The MBD performs verification and validation by testing the proposed mathematical model and algorithms developed to control the Arduino microcontroller, sensors, running motor, pump, and solar energy. The experimental design ensures that the proposed model controls and monitors the automated irrigation system to get feedback from sensors, water levels and activate the watering motor automatically ON/OFF.
* The five layers mentioned in section 2.1 are not shown in fig 1. Is the layered architecture bundled in the off-the-shelf agricultural kit?
Thank you for the helpful comment. Yes, the five layers are embedded in agriculture kit. We updated the text to make it clear and obvious.
The proposed agriculture architecture uses Libelium Smart Agriculture Kit, including various agricultural sensors [23]. The Agriculture Kit consists of five layers: data source layer, data collection layer, data transmission layer, data processing layer, and data viewing layer.
*The three wireless communication protocols mentioned -- are all of them used in the deployment? If yes, why all of them simultaneously? Please justify. If not, what exactly is used in the deployment?
Thank you for the helpful comment. No, the Libelium Agriculture Kit provided three types of communication channels (LoRa, WIFI, and ZigBee). The proposed sytem is used the LoRa communication channel. We updated the text to make it clear and obvious.
This experiment configured each sensor node to send a frame of collected data to the data processing layer for every 15 minutes approximately through the LoRa communication channel.
* In general, this research area can be focused on three aspects -- communication protocol design, data analytics and architecture design that facilitates both efficient communication and data analytics. Please concentrate on one.
Thank you for the helpful comment. Yes, we are focusing on architecture design and data analytics because we needed reliable data to test the proposed architecture efficiently. Please refer to the updated version.
* You have missed a number of relevant works. Please position your work by seeking help from the existing literature, e.g.,
Nawandar et al., "IoT based low cost and intelligent module for smart irrigation system",
Benyezza et al., "Smart Irrigation System Based Thingspeak and Arduino"
Krishnan et al., "Fuzzy Logic based Smart Irrigation System using Internet of Things"
Thank you for the exciting references that indeed enrich the relevant work. We used the first two references related to the proposed system.
Please refer to the updated version.
* For prediction, there exists a number of existing methods as well, e.g.,
Sarkar et al., "VSF: An Energy-Efficient Sensing Framework using Virtual Sensors". Please position your proposal accordingly.
Thank you for the exciting references that indeed enrich the relevant work. We used the first two references related to the proposed system.
Please refer to the updated version.
* Some attention should also be paid to written English.
Thank you for the helpful comment. Done, a native English speaker has reviewed and proofread the article language. Please refer to the updated version.

Reviewer 2 Report
This paper deals with a real problem in agriculture. Motivation is clear. Paper mentions many related works, however the presentation should be improved. Authors present related works at the introduction and only at the end discuss their problems and limitations as a whole. I think that authors should classify related works and discuss problems and advantages at the end of each category comparing them with the proposed approach. Also, authors should discuss clearly their contribution comparing their proposed method with related works at the introduction. Authors implement their proposed system using Libelium Smart Agriculture Vertical Kit. Authors configure suitably their equipment and test it with a rather simple scenario. I cannot see the innovation here except from the efficient usage of their equipment.
Author Response
Manuscript ID: computers-1433128
Experimental and Mathematical Models for Real-time Monitoring and Auto Watering Using IoT Architecture
Dear Editor
Dear respected reviewers
I have revised my article according to the editors' and reviewers' comments without any exception. However, below is my detailed response to the received comments. I would like to take this opportunity to thank the editorial office members and the reviewers for their valued comments and feedback, which enriched the article.
Note: the remarks of the reviewers are colored black, while the authors' respond is highlighted in blue
Best regards
I would like to take this opportunity to thank the reviewers for the valued comments and feedback, which enriched the article.
Thank you for the helpful comment. The manuscript is enhanced and updated according to the reviewer’s comment. Please refer to the new version.
Reviwer2:
This paper deals with a real problem in agriculture. Motivation is clear. Paper mentions many related works, however the presentation should be improved.
Thank you for the helpful comment. The presentation of the manuscript is enhanced and updated based on the reviewer's comment. Please refer to the updated version.
Authors present related works at the introduction and only at the end discuss their problems and limitations as a whole. I think that authors should classify related works and discuss problems and advantages at the end of each category comparing them with the proposed approach.
Thank you for the helpful comment. The related work section is added. Please refer to the updated version.
Also, authors should discuss clearly their contribution comparing their proposed method with related works at the introduction. Authors implement their proposed system using Libelium Smart Agriculture Vertical Kit. Authors configure suitably their equipment and test it with a rather simple scenario. I cannot see the innovation here except from the efficient usage of their equipment.
Thank you for the helpful comment. Done, the manuscript is updated based on the reviewer comment. The contributions of this work are:
This work suggested an automatic irrigation method based on a developed mathematical model derived according to the nature of the land and climatic conditions such as temperature and humidity. The proposed model can be easily and quickly changed to meet any changes in climatic conditions.
Also, the proposed model helps to manage and monitor plants needs in an efficient manner. The use of sensors helps to use water efficiently and reduce water-consuming and needed energy for irrigation. Reduce the need for labor to turn the motor ON and OFF, controlled by the automated irrigation system based on renewable energy.
Most existing systems require a connection to the Internet and external data storage to manage and control the plant's needs. The proposed method helps manage and control the plants' needs automatically without the need for the Internet. It is embedded in the field and can easily update it for any new conditions.

Reviewer 3 Report
This paper proposed an autonomous sensor-enabled architecture using wireless sensors that support real-time monitoring of plants with regard to soil moisture and irrigation in order to minimize water waste and maximize the growth rates of plants. Also, three mathematical regression models were developed to predict future agricultural activities.Suggestions for improvements:
- you mentioned in your paper that "This experiment configured each sensor node to send a frame of collected data to the data processing layer for every 15 minutes approximately."
Was that number chosen arbitrarily or motivated by some other research?
- you stated that "The second node (node 2 - Figure 4. b) was watered manually once per day." with the same amount of water as in node 1?
Minor fixes:
- proofread paper
- no comma before etc.
- fix sentence: "Water is a valuable resource that must preserve."
- fix sentence: "decide whether the plant is needed water or not"
- fix grammar in abstract
- fix "Ashton k. [12] described "
- fix sentence: "This paper proposes IOT embedded system for auto watering ..."
- fix headings, e.g. "2.1. System Set-up and instalation" or "2. 2. Experemintal Set-up and instalation"
- use consistent line spacing
- fix "real-Time monitoring"
- eq. 1, 2, 3, 4, 5 are in bad resolution, fix its quality
- fix sentence: "solve this issue, so safely can use the relay without any problems"
- fix sentence: " to ensure no irregularities in the plant’s environmental conditions"
- fix sentence: "defined as in question 2"
- fix missing part in sentence: "where yi is the experimental data and is the mean of the experimental data"
- use subscript in yi, fi, yij, dij etc.
- fix: "question 4", "question 5"
- fix: "the humidity level in the plant pot (node 2) is reached a high"
- fix: "However, based on the good conditions of the two plants, then the efficient level of soil humidity is between 25% to 35%"
- fix "0 and 0.24"
- fix space in "Y=XB +B0"
- fix casing in "(b). water amount and irrigation time"
- use consistent apostrophes
- add space in "are less in model1 (5.97) than the other models 2"
- use consistent terminology, e.g. data set or dataset
- fix "dif-ferent", "pa-rameters", "al-lows" and other hyphens in conclusions section
Author Response
Manuscript ID: computers-1433128
Experimental and Mathematical Models for Real-time Monitoring and Auto Watering Using IoT Architecture
Dear Editor
Dear respected reviewers
I have revised my article according to the editors' and reviewers' comments without any exception. However, below is my detailed response to the received comments. I would like to take this opportunity to thank the editorial office members and the reviewers for their valued comments and feedback, which enriched the article.
Note: the remarks of the reviewers are colored black, while the authors' respond is highlighted in blue
Best regards
I would like to take this opportunity to thank the reviewer for the valued comments and feedback, which enriched the article.
Reviewer 3:
This paper proposed an autonomous sensor-enabled architecture using wireless sensors that support real-time monitoring of plants with regard to soil moisture and irrigation in order to minimize water waste and maximize the growth rates of plants. Also, three mathematical regression models were developed to predict future agricultural activities.
Thank you for the helpful comment. The manuscript is enhanced and updated according to the reviewer’s comment. Please refer to the new version.
Suggestions for improvements:
- you mentioned in your paper that "This experiment configured each sensor node to send a frame of collected data to the data processing layer for every 15 minutes approximately."
Was that number chosen arbitrarily or motivated by some other research?
Thank you for the helpful comment. The sending time through the sensor is configured to 15 minutes based on the model computation mentioned in other researches and experimental test, such as Yasmin et al. [xx].
Yasmin R, Mikhaylov K, Pouttu A. LoRaWAN for Smart Campus: Deployment and Long-Term Operation Analysis. Sensors. 2020 Jan;20(23):6721.
Choosing 15 minutes will help reduce the power consumption and save its associated charged battery using an external solar panel. Also, it enables tracking the environment parameters with a sufficient level of accuracy.
- you stated that "The second node (node 2 - Figure 4. b) was watered manually once per day." with the same amount of water as in node 1?
Thank you for the helpful comment. The second node was watered manually once per day when needed. Yes, the both gives the same amount of water half-liter (0.5L) if needed (sensor-based, manual-based).
The manuscript is updated based on the reviewer’s comment. Please refer to the updated version.
Minor fixes:
Thank you for your efforts and helpful comments. The grammatical errors and other issues mentioned below were fixed. Done, a native English speaker has reviewed and proofread the article language. Please refer to the updated version.
- proofread paper
- no comma before etc.
- fix sentence: "Water is a valuable resource that must preserve."
- fix sentence: "decide whether the plant is needed water or not"
- fix grammar in abstract
- fix "Ashton k. [12] described "
- fix sentence: "This paper proposes IOT embedded system for auto watering ..."
- fix headings, e.g. "2.1. System Set-up and instalation" or "2. 2. Experemintal Set-up and instalation"
- use consistent line spacing
- fix "real-Time monitoring"
- eq. 1, 2, 3, 4, 5 are in bad resolution, fix its quality
All the equations were rewritten as presented below. Please refer to the updated version.
(1)
|
(2) |
|
(3) |
|
|
(4) |
|
RMSE |
(5) |
Thank you for your time and helpful comments. We fixed all the grammatical errors and other issues mentioned below. Please refer to the updated version.
- fix sentence: "solve this issue, so safely can use the relay without any problems"
- fix sentence: " to ensure no irregularities in the plant’s environmental conditions"
- fix sentence: "defined as in question 2"
- fix missing part in sentence: "where yi is the experimental data and is the mean of the experimental data"
- use subscript in yi, fi, yij, dij etc.
- fix: "question 4", "question 5"
- fix: "the humidity level in the plant pot (node 2) is reached a high"
- fix: "However, based on the good conditions of the two plants, then the efficient level of soil humidity is between 25% to 35%"
- fix "0 and 0.24"
- fix space in "Y=XB +B0"
- fix casing in "(b). water amount and irrigation time"
- use consistent apostrophes
- add space in "are less in model1 (5.97) than the other models 2"
- use consistent terminology, e.g. data set or dataset
- fix "dif-ferent", "pa-rameters", "al-lows" and other hyphens in conclusions section

Reviewer 4 Report
The authors describe an automated irrigation system with some measurements and mathematical models. The paper is interesting to read but the main idea behind the paper has been shown previously. I think it is worth publishing after improving the presentation and introduction but perhaps in a more specific journal suck as electronics MDPI
Author Response
Reviewer 4
The authors describe an automated irrigation system with some measurements and mathematical models. The paper is interesting to read but the main idea behind the paper has been shown previously. I think it is worth publishing after improving the presentation and introduction but perhaps in a more specific journal suck as electronics MDPI.
Thank you for the helpful comment. The manuscript is enhanced according to the reviewer’s comment. The new idea is to control the irrigation automatically and reduce water consumption at a low cost, controlled by dynamic mathematical models that are easily updated on the field conditions and demand. Please refer to the new version.
The following text was added.
In the introduction:
The need for efficient management irrigation systems has become crucial in many regions worldwide due to the scarcity of water resources because of the changes in climatic conditions, high atmosphere temperature, and the negative impact of human behavior on the environment. Availability of water has become a global problem affecting many countries, especially in remote and desert areas. Oman is one of the countries with large desert areas that lack potable water sources, and the rise in temperatures also leads to the rapid loss of water from the land. Therefore, there is a need for an efficient irrigation system that works automatically to improve irrigation operations, reduce water consumption and reduce energy costs. The purpose of watering is to give the plants the right amount of water to ensure ideal growth. Optimal irrigation management aims to determine the timing and quantity of water suitable for irrigation to achieve the most significant effectiveness. Developments in industry tools, information technology, and communication have helped innovate irrigation methods that consume less water than manual and old technologies [1]. Therefore, intelligent irrigation methods lead to less water consumption and reduce the field's excess water, which leads to better crop production [2]. Finding improved techniques that improve water use efficiency and lower energy usage has become an affluent research area. Developing an autonomous architecture is considered an ideal approach for processing and analyzing the sensed data for supporting real-time monitoring of agricultural parameters. It also ensures interoperability among heterogeneous sensing data streams to support large-scale agricultural analytics [3]. In recent years, precision agriculture has received considerable concern due to the increasing demand for food production with high-quality crops, minimum cost, and reducing the effects of environmental pollution. Wireless sensor network technologies are utilized for providing solutions in the agricultural domain. It aims to provide an optimal tool for collecting, processing, managing, and analyzing the relevant agricultural information and farming activities [4]. The main advantage of these technologies is their ability to create a network of enabled devices (i.e., sensors) that can capture environmental parameters related to agriculture fields and transmit them to the predefined application for further processing and analysis [5]. However, many plantations' attributes such as soil types, fertilizer processes, water requirements, and weather conditions in agriculture fields have different needs and considerations [6]. Many researchers have discussed the need to develop a self-watering mechanism to increase the efficiency of farming systems and reduce the percentage of discharged water.
This work suggested an automatic irrigation method based on a developed mathematical model derived according to the nature of the land and climatic conditions such as temperature and humidity. The proposed model can be easily and quickly changed to meet any changes in climatic conditions.
Also, the proposed model helps to manage and monitor plants needs in an efficient manner. The use of sensors helps to use water efficiently and reduce water-consuming and needed energy for irrigation. Reduce the need for labor to turn the motor ON and OFF, controlled by the automated irrigation system based on renewable energy.
Most existing systems require a connection to the Internet and external data storage to manage and control the plant's needs. The proposed method helps manage and control the plants' needs automatically without the need for the Internet. It is embedded in the field and can easily update it for any new conditions.
Also, the below text was added
- Related Work
Many researchers have proposed autonomous methods for watering plants based on mathematical models derived by machine learning methods.
Nawandar et al. [23] proposed a low-cost intelligent irrigation system using a neural network method for determining the sensor input based on the irrigation schedule for efficient irrigation. The proposed devices offered several facilities, such as irrigation schedule estimation, decision making, and remote data monitoring. Sarkar et al. [24] developed a virtual sensing framework (VSF), which helped reduce the network's data traffic and transmission. They deployed a cross-correlation method for predicting multiple consecutive sensed data and achieved an accuracy of 98%. Benyezza et al. [25] developed an automated irrigation embedded system based on Arduino for optimizing water use and monitoring the field.
- 3. Materials and Methods
This section describes the proposed IoT architecture based on Experimental and Mathematical Models for auto watering and real-time monitoring of heterogeneous sensing agricultural parameters.
This manuscript deployed a Model-based design (MBD) and experimental research methods for developing an embedded automatic irrigation control system. The MBD performs verification and validation by testing the proposed mathematical model and algorithms developed to control the Arduino microcontroller, sensors, running motor, pump, and solar energy. The experimental design ensures that the proposed model controls and monitors the automated irrigation system to get feedback from sensors, water levels and activate the watering motor automatically ON/OFF.
4.3 Mathematical Regression Models
Comparing the results with other researchers is one of the important ways to verify the effectiveness of the extracted results compared to other studies. The results should be compared under the same conditions to give credibility to the results extracted. But one of the most significant obstacles that we face when carrying out the comparative study is the disparity of the work environment and the conditions of input and outputs. Therefore, we often try to find some common parameters and compare them based on them.
Table No. 6 presents the results of the proposed methods compared to some studies, which show that most of the proposed methods and these studies have achieved high accuracy. They significantly reduce the percentage of water and energy consumption, which indicates the success of the current experiments. The proposed models reduce water consumption about 50% to 65% compared to 30-60% in [14] and 12.5% in [23]. In addition, another study [24] reduced energy consumption up to 69% compared with the proposed models that reduce the energy by using a solar panel to charge the battery. The Table
|
|
Model1 |
Model2 |
Model3 |
[23] |
[14] |
[24] |
|
|
R2 |
0.73 |
0.64 |
0.65 |
0.98 |
- |
- |
|
|
R2_adj |
0.73 |
0.64 |
0.65 |
- |
- |
- |
|
|
MSE |
5.97 |
7.85 |
0.132 |
0.06 |
- |
0.22 |
|
|
RMSE |
2.44 |
2.80 |
0.363 |
0.77 |
- |
0.47 |
|
|
Saving Water |
50-75% |
50-75% |
50-75% |
12.5% |
30-60% |
- |
|
|
Saving Energy |
Yes |
Yes |
Yes |
- |
- |
69% |
|
Table 6. The comparison results of the proposed methods with other studies.

Round 2
Reviewer 2 Report
After the revision my main concerns remain. I feel that this paper is a case study with small contribution. Also, I'd like to see a comparison of the proposed system with other related systems in the literature.
Author Response
Reviewer 2:
After the revision, my main concerns remain. I feel that this paper is a case study with a small contribution. Also, I'd like to see a comparison of the proposed system with other related systems in the literature.
Thank you for the helpful comment. The manuscript is enhanced according to the reviewer’s comment. Please refer to the new version.
Comparing the results with other researchers is one of the important ways to verify the effectiveness of the extracted results compared to other studies. The results should be compared under the same conditions to give credibility to the results extracted. But one of the most significant obstacles that we face when carrying out the comparative study is the disparity of the work environment and the conditions of input and outputs. Therefore, we often try to find some common parameters and compare them based on them.
Table No. 6 presents the results of the proposed methods compared to some studies, which show that most of the proposed methods and these studies have achieved high accuracy. They significantly reduce the percentage of water and energy consumption, which indicates the success of the current experiments. The proposed models reduce water consumption about 50% to 75% compared to 30%-60% in [14] and 12.5% in [23]. In addition, another study [24] reduced energy consumption up to 69% compared with the proposed models that reduce the energy by using a solar panel to charge the battery.

Reviewer 3 Report
The paper did not address all the minor fixes. However, it is publishable in this form.
Round 3
Reviewer 2 Report
After the revision I feel that this paper has been improved significantly.